# Transcriptome Analysis of Genes Involved in Cold Hardiness of Peach Tree (*Prunus persica*) Shoots during Cold Acclimation and Deacclimation

**DOI:** 10.3390/genes11060611

**Published:** 2020-06-01

**Authors:** Duk Jun Yu, Sung Hoon Jun, Junhyung Park, Jung Hyun Kwon, Hee Jae Lee

**Affiliations:** 1Department of Plant Science, Seoul National University, Seoul 08826, Korea; dukjunyu@snu.ac.kr (D.J.Y.); 1128hoon@hanmail.net (S.H.J.); trecea@snu.ac.kr (J.P.); kwon1101@korea.kr (J.H.K.); 2Research Institute of Agriculture and Life Sciences, Seoul National University, Seoul 08826, Korea; 3Fruit Research Division, National Institute of Horticultural and Herbal Science, Wanju 55365, Korea

**Keywords:** cold hardiness, peach tree, transcriptome, cold acclimation, deacclimation

## Abstract

We analyzed the transcriptomes in the shoots of five-year-old ‘Soomee’ peach trees (*Prunus persica*) during cold acclimation (CA), from early CA (end of October) to late CA (middle of January), and deacclimation (DA), from late CA to late DA (middle of March), to identify the genes involved in cold hardiness. Cold hardiness of the shoots increased from early to late CA, but decreased from late CA to late DA, as indicated by decreased and increased the median lethal temperature (LT_50_), respectively. Transcriptome analysis identified 17,208 assembled transcripts during all three stages. In total, 1891 and 3008 transcripts were differentially expressed with a |fold change| > 2 (*p* < 0.05) between early and late CA, and between late CA and late DA, respectively. Among them, 1522 and 2830, respectively, were functionally annotated with gene ontology (GO) terms having a greater proportion of differentially expressed genes (DEGs) associated with molecular function than biological process or cellular component categories. The biochemical pathways best represented both periods from early to late CA and from late CA to late DA were ‘metabolic pathway’ and ‘biosynthesis of secondary metabolites’. We validated these transcriptomic results by performing reverse transcription quantitative polymerase chain reaction on the selected DEGs showing significant fold changes. The relative expressions of the selected DEGs were closely related to the LT_50_ values of the peach tree shoots: ‘Soomee’ shoots exhibited higher relative expressions of the selected DEGs than shoots of the less cold-hardy ‘Odoroki’ peach trees. Irrespective of the cultivar, the relative expressions of the DEGs that were up- and then down-regulated during CA, from early to late CA, and DA, from late CA to late DA, were more closely correlated with cold hardiness than those of the DEGs that were down- and then up-regulated. Therefore, our results suggest that the significantly up- and then down-regulated DEGs are associated with cold hardiness in peach tree shoots. These DEGs, including *early light-induced protein 1*, *chloroplastic*, *14-kDa proline-rich protein DC2.15*, *glutamate dehydrogenase 2*, and *triacylglycerol lipase 2,* could be candidate genes to determine cold hardiness.

## 1. Introduction

Cold, including freezing, is a major environmental stress for deciduous fruit trees, limiting their geographical distribution, growth, and productivity. To survive harsh winters and regrow the following spring, deciduous fruit trees change their cold hardiness throughout the year via cold acclimation (CA) and deacclimation (DA) [1,2,3]. CA is a complex process including the modification of membrane lipid composition, the increases in compatible solutes, the enhancement of antioxidative mechanisms, and the biosynthesis of protective proteins [1,4]. During CA, cold stress modifies the expression of various genes and their related metabolisms with consequent effects on many biological functions. The major changes that occur during CA are mostly reversed during DA [3].

Enzymes and proteins that directly protect cells against abiotic stresses, including cold stress, are important for the survival of plants. Stress responses are regulated by multiple genes, and high-throughput sequencing and functional genomics tools can help unravel the molecular mechanisms involved. In particular, transcriptome sequencing analyses can provide a framework dataset for the determination of metabolic pathways, the clarification of gene expression patterns, and the mining of new genes. Gene expression profiling has been attempted to identify genes involved in the responses to low temperatures in deciduous fruit trees including grape (*Vitis* spp.) [5] and peach (*Prunus persica*) [6]. In peach trees, the expression of genes, including cold-response genes, has been investigated [6,7,8,9] since the peach genome was sequenced [10]. However, most of these studies have focused on the expression of known genes in specific tissues or on the changes in their expression when exposed to low temperatures [7,8,9]. For instance, Artlip et al. [7] verified the influence of low temperature and dehydration on the expression levels of *C-repeat-binding factor* (*CBF*) and *dehydration-responsive element-binding factor 2* (*DREB2*) genes in peach leaf and bark tissues. Bassett et al. [8] differentiated the dehydrin members through their expression analysis in the bark and leaves of peach trees subjected to low temperatures and/or dehydration. Jiao et al. [9] characterized the transcriptomes of peach stigmas and identified differentially expressed genes (DEGs) following the treatment at −2 °C for 4 and 26 h. However, these transcriptome analyses have been performed under specific temperature conditions.

To obtain overall view of the metabolic changes related to cold hardiness, we analyzed the transcriptomes of ‘Soomee’ peach tree shoots during CA, from early to late CA, and DA, from late CA to late DA, under field conditions. Transcriptome profiling from late CA to late DA provides information about the resumption of growth and development in the spring and can also be useful for filtering out the transcripts that are differentially expressed from early to late CA but irrelevant to cold hardiness. Since genes that are significantly regulated during these periods might be related to cold hardiness, we compared their expression patterns with those in the shoots of ‘Odoroki’ peach trees, which are less cold-hardy than ‘Soomee’, to validate their involvement in cold hardiness.

## 2. Materials and Methods

### 2.1. Plant Materials

Five-year-old ‘Soomee’ and ‘Odoroki’ peach trees grown under field conditions in the experimental orchard of the National Institute of Horticultural and Herbal Science (35°82′ N, 127°02′ E), Rural Development Administration, Wanju, Republic of Korea, were used in this study. Bud-attached shoots from three different ‘Soomee’ and ‘Odoroki’ peach trees each were separately collected at the end of October, middle of January, and middle of March, representing early CA, late CA, and late DA, respectively, to provide three biological replicates. Six and three shoots from each tree were used for cold hardiness determination and transcriptome analysis, respectively, at each physiological stage.

### 2.2. Cold Hardiness Determination

The cold hardiness of the peach tree shoots was determined using an electrolyte leakage analysis following the methods of Pagter et al. [11] and Lee et al. [12] with slight modifications. To avoid the damage caused by bud separation from the shoots, bud-attached shoots were cut into 8-cm pieces and randomly divided into six groups. Four groups out of six were incubated in a programmable bath circulator (RW-2040G; Jeio Tech, Seoul, Republic of Korea), cooled at a rate of −2 °C/h until they reached each target temperature, and maintained at the target temperature for 2 h. Four target temperatures ranging from −5 °C to −35 °C were selected. The incubated samples were thawed at 0 °C, and all temperatures were recorded every second using a data logger (CR-1000 M; Campbell Scientific, Inc., Logan, UT, USA) with a copper–constantan thermocouple. For comparison, the other two groups were separately incubated in a refrigerator at 4 °C and in a freezer at −80 °C.

Following the freezing treatment, the internodes from the bud-attached shoots were cut into 1-cm pieces, and five pieces of each sample were shaken in a 50-mL tube containing 10 mL distilled water at 125 rpm on an orbital shaker (Supertech Orbital Shaker; SeouLin Bioscience, Seoul, Republic of Korea) at room temperature for 24 h. The electrical conductivity (EC) of each aliquot was then measured using an EC meter (Orion Star A215; Thermo Fisher Scientific, Waltham, MA, USA). After autoclaving of the samples at 120 °C for 30 min, the EC was measured again. The percentage of injury was calculated as described by Arora et al. [13] and adjusted using the equation reported by Yu et al. [14]. The median lethal temperature (LT_50_) was calculated using the Gompertz function and used as a measure of cold hardiness. The adjusted injury values of each cultivar at each physiological stage were resampled 30 times as described by Arora et al. [15] to obtain efficient LT_50_ estimates without repeating the entire experiment.

### 2.3. Total RNA Extraction

The internodes from the bud-attached shoots were immediately frozen in liquid nitrogen upon the shoot collection and stored at −80 °C. Total RNA was extracted from the shoot internodes using cetyltrimethylammonium bromide buffer as described by Gambino et al. [16] with slight modifications. A 900 μL aliquot of the extraction buffer was preheated to 65 °C and added to a 2-mL microcentrifuge tube containing 100 mg of shoot powder, mixed thoroughly, and incubated at 65 °C for 10 min. An equal volume of chloroform:isoamyl alcohol (24:1, *v/v*) was added, vortexed for 5 s, and centrifuged at 11,000× *g* at 4 °C for 10 min. The 750-μL supernatant was recovered and again mixed with an equal volume of the chloroform:isoamyl alcohol. The 600-μL supernatant was transferred to a new 2-mL tube and an equal volume of 6-M LiCl was added. The mixture was incubated on ice for 30 min and centrifuged at 21,000× *g* at 4 °C for 20 min to precipitate the RNA. The pellet was resuspended in 500 μL of preheated (65 °C) SSTE buffer consisting of 0.5% sodium lauryl sulfate, 1 M NaCl, 1 M Tris-HCl (pH 8.0), and 10 mM EDTA with gentle shaking. An equal volume of the chloroform:isoamyl alcohol was added, and the mixture was centrifuged at 11,000× *g* at 4 °C for 10 min. The 400-μL supernatant was mixed with 280 μL cold isopropanol and centrifuged at 21,000× *g* at 4 °C for 15 min. The pellet was washed with 1 mL of 70% ethanol, dried, and resuspended in 20 μL of diethyl pyrocarbonate-treated water. Finally, the solution was heated at 65 °C for 5 min to dissolve the RNA.

The quality and purity of the extracted RNA samples were assessed by determining their *A*_260_/*A*_280_ ratios and RNA integrity numbers using a Nanodrop ND1000 spectrophotometer (Thermo Fisher Scientific) and an Agilent 2100 Bioanalyzer (Agilent Technologies, Santa Clara, CA, USA), respectively.

### 2.4. cDNA Library Construction, Sequencing, and Assembly

A total of nine cDNA libraries were constructed from RNA extracted from the ‘Soomee’ peach tree shoots at the three physiological stages using a TruSeq RNA sample prep kit v2 (Illumina, San Diego, CA, USA). They were sequenced using a HiSeq 2500 system (Illumina) to generate 101-bp paired-end reads. The quality of the resulting data was confirmed using the FastQC v0.11.5 program (http://www.bioinformatics.babraham.ac.uk/projects/fastqc/), and unwanted artifacts, including the adaptor sequence, low-quality reads, and short-length reads (<36 bp), were removed from the raw data using the Trimmomatic v0.32 program (http://www.usadellab.org/cms/?page=trimmomatic). The statistical values, including the GC content and Phred quality score 30 (Q30), of the trimmed data were calculated. The trimmed reads were mapped to the reference peach genome (GCF_000346465.2) using the Bowtie2 aligner and the HISAT2 program (https://ccb.jhu.edu/software/hisat2/index.shtml), after which the aligned reads were assembled into transcripts or genes using the StringTie 1.3.3b program (https://ccb.jhu.edu/software/stringtie). The RNA-sequencing data were deposited in the National Center for Biotechnology Information (NCBI) database (accession No. PRJNA587386).

### 2.5. Identification and Functional Annotation of the DEGs

The abundances of the assembled transcripts in each sample were expressed by normalized fragments per kilobase of transcript per million mapped reads (FPKM) values. To effectively identify the DEGs, the assembled transcripts or genes showing zero FPKM in any sample were removed from the analysis. The filtered data were adjusted using the quantile normalization method after taking the log_2_(FPKM + 1) values to reduce the range of the data and evenly distribute the data. Significant DEGs involved in cold hardiness were identified by performing paired comparisons between the normalized FPKM values from early to late CA and between those from late CA to late DA using the Student’s *t*-test at *p* < 0.05. The DEGs were selected based on the significance of fold changes between early and late CA and between late CA and late DA. The fold changes were calculated as the ratios between squared values of normalized FPKM means at each stage. Functional annotation and gene set enrichment analysis were also performed using the DAVID tool (http://david.abcc.ncifcrf.gov) based on their associated gene ontology (GO) terms and Kyoto Encyclopedia of Genes and Genomes (KEGG) database categories (http://www.genome.jp/kegg/).

### 2.6. Reverse Transcription Quantitative Polymerase Chain Reaction (qPCR) Analysis

To validate DEG identification results and analyze their expression patterns, qPCR analyses were performed on selected transcripts showing significant fold changes between early and late CA and between late CA and late DA. Primer sets were designed using NCBI Primer-BLAST, and the duplication of cDNA using these primers was checked by electrophoresis on 1% agar in 0.02 M Tris-acetate (pH 8.0) buffer including 0.5 mM EDTA. The relative expressions of the transcripts were determined using a LightCycler 480 system (Roche Diagnostics, Basel, Switzerland) and an AccuPower Greenstar qPCR master mix (Bioneer, Daejeon, Republic of Korea). The results were standardized to the expression of the gene encoding glyceraldehyde-3-phosphate dehydrogenase, as described by Zifkin et al. [17]. The relative expressions were plotted using SigmaPlot 12 (version 8.0.2; GraphPad Software Inc., San Diego, CA, USA).

### 2.7. Statistical Analysis

Statistically significant differences among the means were determined using the Student’s *t*-test, the modified Fisher’s exact test, or the Duncan’s multiple range test at *p* < 0.05, 0.01, or 0.001 using the R v3.2.2 software package (http://www.r-project.org). The Benjamini–Hochberg method was also applied to determine statistically significant differences at false discovery rate < 0.05 or 0.01.

## 3. Results and Discussion

### 3.1. Changes in Cold Hardiness

The LT_50_ values of the peach tree shoots significantly decreased during CA, from early to late CA, and increased during DA, from late CA to late DA, irrespective of the cultivar (Figure 1). Based on the LT_50_ values, the ‘Soomee’ peach tree shoots were cold-hardier than the ‘Odoroki’ peach tree shoots throughout both periods.

### 3.2. Transcriptome Sequencing Data Statistics

Transcriptome sequencing of the ‘Soomee’ peach tree shoots during CA, from early to late CA, and DA, from late CA to late DA, generated 50,845,192–66,451,463 reads with a GC content of 45.6–46.3% and a Q30 of 93.9–96.3% (Table 1). An average of 5.12–6.68 × 10^9^ bases of sequence were obtained for each sample, and almost all of the total reads were mapped to the reference peach genome (GCF_00346464.2), with a mapping ratio of 96.7–97.2% (Table 1).

### 3.3. DEG Analysis

The transcripts expressed in any shoot samples examined were counted to be 18,344. Following the removal of the transcripts showing zero FPKM in any samples, the numbers of the transcripts identified in all the samples examined were 17,208. Of the transcripts, 1891 and 3008 were differentially expressed between early and late CA, and between late CA and late DA, respectively, with a |fold change| > 2 (*p* < 0.05) (Figure 2). The number of significant DEGs between late CA and late DA was higher than that between early to late CA, which indicates that DA is not merely a reverse of CA, but a separate physiological stage preparing trees for the resumption of growth and development. During these periods, 926 and 1624 transcripts were up-regulated, respectively, while 965 and 1384 transcripts were down-regulated, respectively (Figure 2). The number of DEGs with a |fold change| > 10 (*p* < 0.05) was generally <100 during these periods.

### 3.4. Functional Annotation and GO Term Enrichment of the DEGs during CA, from Early to Late CA 

GO terms have widely been applied to understanding the biological significance of DEGs. Among the 1891 DEGs during CA, from early to late CA, 1522 could be functionally annotated and were classified into functional terms within the three main GO categories (Figure 3A). A relatively large proportion of the functionally annotated DEGs were associated with various molecular function terms (402 DEGs; 26.4%). Fewer DEGs were associated with the biological process (207 DEGs; 13.6%) and cellular component terms (194 DEGs; 12.7%), with a similar proportion of DEGs enriched in these two categories.

In the biological process category, the GO term ‘metabolic process’ represented the most highly enriched group among the DEGs, followed by ‘cell wall organization’ (Figure 3A). The exploration of DEGs associated with the ‘metabolic process’ term may allow the identification of genes involved in altering secondary metabolic pathways during CA, from early to late CA. For examples, *squalene monooxygenase* (LOC18768153), *β-glucosidase 12* (LOC18770931), and *adenylate isopentenyltransferase 3*, *chloroplastic* (LOC18766570) could be included as up-regulated genes, while *isoleucine N-monooxygenase 2* (LOC18775172), *polyphenol oxidase*, *chloroplastic* (LOC18778195), and *gibberellin 2-β-dioxygenase* (LOC18779667) could be included as down-regulated genes. Sanghera et al. [18] reported that changes in the expression of genes in response to cold stress were followed by increases in the levels of hundreds of metabolites, some of which are known to protect against cold stress. ‘Cell wall organization’ is a process that results in the assembly or disassembly of the cell wall or regulates the arrangement of its constituent parts, maintaining the shape of the cell and protecting it from osmotic lysis. CA was previously shown to induce changes in the cell wall polysaccharide composition and in the activities of the cell wall-modifying enzymes [19,20]. Gall et al. [19] reported that two main mechanisms could be highlighted in the responses to abiotic stresses, including cold stress: an increase in the xyloglucan endotransglucosylase/hydrolase and expansin proteins involved in cell wall plasticity, and an increase in cell wall thickness resulting from the reinforcement of the secondary wall with hemicellulose and lignin. In the present study, genes encoding xyloglucan endotransglucosylase/hydrolase were particularly significantly regulated in the ‘Soomee’ peach tree shoots during CA, from early to late CA (Appendix A).

In the molecular function category, the enriched DEGs were mainly associated with the ‘metal ion binding’ and ‘transcription factor activity, sequence-specific DNA binding’ GO terms (Figure 3A), presumably representing genes involved in signal transduction pathways during CA, from early to late CA. Many proteins require bound metals to achieve their functions, and the metal-binding proteins are associated with a variety of cellular functions, including cell signaling [21]. Transcription factors (or sequence-specific DNA binding factors) are proteins that control the transcription of genetic information from DNA into mRNA and play vital regulatory roles in abiotic stress responses in plants [22,23]. The CBF transcription factors (also known as DREB1s), for example, are members of the AP2/EREBP (APETALA 2/ethylene-responsive element-binding factor) family, and mediate freezing tolerance of plants by binding to CBF cis-elements (5′-A/GCCGAC-3′) located in the promotors of cold-regulated genes and by up-regulating their expression [22,24,25]. In the present study, five *DREB* genes, *DREB1A* (LOC18778067), *DREB1E* (LOC18776669), *DREB2C* (LOC18785445), *DREB2D* (LOC18789405), and *DREB3* (LOC18793847), were identified and significantly differentially regulated during CA, from early to late CA, and DA, from late CA to late DA. *DREB1E*, *DREB2C*, and *DREB**2D* were significantly up- and down-regulated during CA, from early to late CA, and DA, from late CA to late DA, respectively, while *DREB1A* and *DREB**3* were significantly down-regulated during both periods (Appendix A).

In the cellular component category, the GO term ‘nucleus’ was dominant among the DEGs (Figure 3A); other enriched terms included ‘extracellular region’, ‘cell wall’, and ‘apoplast’. The cold tolerance of plants often depends on the mechanical properties of the cell wall [19]. During drastic cold exposure, freezing can induce extracellular ice formation, which leads to cell dehydration and subsequent collapse. Cell wall rigidity might determine the resistance of cells to this freezing-induced dehydration. CA can induce changes in lignin content and composition, thus preventing freezing damage and cell collapse [19,20], and may also induce changes in the composition of the cell wall polysaccharides and in the activities of cell wall-modifying enzymes [19].

### 3.5. Functional Annotation and GO Term Enrichment of the DEGs during DA, from Late CA to Late DA

The GO term enrichment of the DEGs identified in the ‘Soomee’ peach tree shoots were markedly different between CA, from early to late CA, and DA, from late CA to late DA (Figure 3). More functionally annotated DEGs were identified between late CA and late DA than between early and late CA, and the DEGs seen between late CA and late DA were classified into more terms within each GO category. During DA, from late CA to late DA, 2830 DEGs were functionally annotated and classified into functional terms within the three main GO categories (Figure 3B). The proportion of DEGs associated with the biological process terms (402 DEGs; 14.2%) was lower than those of DEGs associated with the molecular function (1161 DEGs; 41.0%) and cellular component terms (916 DEGs; 32.3%).

In the biological process category, the enriched DEGs were mainly associated with the ‘carbohydrate metabolic process’ and ‘transmembrane transport’ GO terms during DA, from late CA to late DA (Figure 3B), unlike during CA, from early to late CA (Figure 3A). The ‘carbohydrate metabolic process’ term is associated with the chemical reactions and pathways involved in carbohydrate metabolism, including the formation of carbohydrate derivatives. ‘Transmembrane transport’ is a process by which a solute is transported across a lipid bilayer. The GO terms ‘ATP binding’ and ‘integral component of membrane’ were dominant in the molecular function and cellular component categories, respectively, during DA, from late CA to late DA (Figure 3B). These GO terms suggest that the expression of genes involved in the metabolic processes preparing for the resumption of growth and development is induced as cold hardiness decreases during DA, from late CA to late DA. Since the genes encoding 1-aminocyclopropane-1-carboxylate oxidase 1 (LOC18781368), leucoanthocyanidin reductase (LOC18789589), peroxidases (LOC18773443, LOC18769960, and LOC18772634), endoglucanase 9 (LOC18768088), and phosphoenolpyruvate carboxykinase (ATP) (LOC18775166) were significantly up-regulated only from late CA to late DA, they might be involved in the metabolic processes responsible for the resumption of growth and development.

### 3.6. KEGG Category Enrichment of the DEGs

According to the results of the biochemical pathways analysis based on the KEGG categories, 930 DEGs detected during CA, from early to late CA, were assigned to 110 KEGG pathways, while 1467 DEGs detected during DA, from late CA to late DA, were assigned to 116 KEGG pathways (Figure 4). The most represented pathways were ‘metabolic pathways (MapID 01100)’ (174 and 286 members during CA and DA, respectively) and ‘biosynthesis of secondary metabolites (MapID 01110)’ (120 and 185 members during CA and DA, respectively). The genes encoding squalene monooxygenase (LOC18768153) and glutamate dehydrogenase 2 (LOC18785018) associated with ‘metabolic pathways’ were greatly up- and then down-regulated during CA, from early to late CA, and DA, from late CA to late DA, respectively, while the gene encoding phosphoenolpyruvate carboxykinase (ATP) (LOC18775166) was significantly down- and then up-regulated during CA, from early to late CA, and DA, from late CA to late DA, respectively (Table 2 and Table 3). Squalene monooxygenase, glutamate dehydrogenase, and phosphoenolpyruvate carboxykinase (ATP) are involved in sterol biosynthesis, nitrogen or glutamate metabolism, and the decarboxylation of oxaloacetate to phosphoenolpyruvate in the gluconeogenesis pathway, respectively [26,27,28]. These KEGG database annotations associated with the biochemical pathways provide a valuable resource for understanding the complex functions and utilities of the biological systems involved in the cold stress response in peach tree shoots.

### 3.7. Validation of In Silico DEG Data Using qPCR

Although the number of DEGs detected in the response to cold stress differed between CA, from early to late CA, and DA, from late CA to late DA, most were identified during both periods. Considering the changes in cold hardiness, these common DEGs might play more important roles in determining the cold hardiness level in the ‘Soomee’ peach tree shoots than the other identified DEGs.

To validate the DEGs and transcriptome data from the ‘Soomee’ peach tree shoots, we selected 18 common DEGs showing a significant fold change during both periods, and analyzed their expression patterns using qPCR. The selected DEGs consisted of nine genes up- and then down-regulated, and nine genes down- and then up-regulated during CA, from early to late CA, and DA, from late CA to late DA, respectively, and the specific primer sets used to analyze their expression are listed in Table 4. Most of the selected genes were differently expressed during CA, from early to late CA, and DA, from late CA to late DA (Figure 5), and their expressions were closely correlated with the in silico FPKM values of the selected DEGs or seasonal changes in the cold hardiness (LT_50_) of the ‘Soomee’ peach tree shoots (Table 2). According to the correlation coefficients (*r*) between the relative gene expressions of each DEG and LT_50_ values, the relative expressions of the DEGs up-regulated during CA, from early to late CA, and down-regulated during DA, from late CA to late DA, were more closely correlated with cold hardiness than those of the DEGs down-regulated during CA, from early to late CA, and up-regulated during DA, from late CA to late DA (Table 2). These results suggest that the DEGs up-regulated during CA, from early to late CA, and down-regulated during DA, from late CA to late DA, are more influential determinants of the cold-hardiness level than those displaying the opposite pattern in the ‘Soomee’ peach tree shoots.

We also compared the relative expressions of the nine DEGs that were up-regulated during CA, from early to late CA, and down-regulated during DA, from late CA to late DA, in the ‘Soomee’ peach tree shoots with those in the less cold-hardy ‘Odoroki’ peach tree shoots (Figure 1). Irrespective of the cultivar, the relative expressions of most of these DEGs were significantly up- and then down-regulated during CA, from early to late CA, and DA, from late CA to late DA, respectively (Figure 6). The ‘Soomee’ shoots showed higher relative expression levels of these DEGs than the ‘Odoroki’ shoots, with the exception of the DEG encoding non-symbiotic hemoglobin (LOC18766910) and tonoplast dicarboxylate transporter (LOC18780397), which were higher in ‘Odoroki’ than in ‘Soomee’ during both periods (Figure 6).

### 3.8. Selection of the DEGs Involved in the Cold Hardiness of Peach Tree Shoots

The correlation of DEG expression levels with the seasonal changes in cold hardiness that occur during CA, from early to late CA, and DA, from late CA to late DA, suggests that the DEGs common to these two periods with higher fold changes might be closely involved in determining cold hardiness in the peach tree shoots. However, a higher fold change does not necessarily imply that a DEG has a major influence on the determination of cold hardiness, since DEGs with low expression volumes (geometric means of FPKM values) may not play a crucial role in the response to cold stress despite their high fold change.

Among the significant common DEGs restricted with false discovery rate < 0.05 [29], 322 DEGs were significantly up- and then down-regulated during CA, from early to late CA, and DA, from late CA to late DA, respectively, while 327 DEGs were significantly down- and then up-regulated. Top 20 common DEGs up-regulated during CA, from early to late CA, and down-regulated during DA, from late CA to late DA, were selected based on their fold changes and expression volumes (Table 3). Most of these common DEGs are functional genes encoding enzymes and metabolic proteins, which can directly protect cells against stresses. The expression levels of the common DEGs exhibited a higher correlation with the cold hardiness level (Table 3), suggesting that they influence cold hardiness in peach tree shoots. The DEGs included *early light-induced protein 1*, *chloroplastic*, *14-kDa proline-rich protein DC2.15*, *glutamate dehydrogenase 2*, and *triacylglycerol lipase 2*. However, their roles in enhancing cold hardiness require further elucidation in future studies. Cold stress accompanies dehydration of the tissues of woody plants during CA. Dehydration can significantly up-regulate dehydration-responsive genes, including *low-temperature-induced 65 kDa protein* (LOC18786412) [30,31], *late embryogenesis-abundant* (*LEA*) *protein 29* (LOC18788434) [32], *dehydrin Xero 2* (LOC18769991) [33], and *desiccation-related protein PCC13-62* (LOC18789656) [34]. In the present study, however, these genes were not selected as top-ranking DEGs despite their particularly high fold changes and expression volumes during DA, from late CA to late DA. According to the in silico data, the dehydration-responsive genes mentioned above were up-regulated at early CA, maintained at their high expression levels from early to late CA, and were then greatly down-regulated from late CA to late DA (Appendix A). These results suggest that dehydration-responsive genes are involved in the cold hardiness of peach tree shoots in a different manner to the selected common DEGs.

Numerous studies on functional genes have mainly focused on those encoding enzymes and proteins that are protective during exposure to a given stress [8,12,30,32,35,36]. Such stress-responsive functional genes include those that regulate the accumulation of compatible solutes, passive and active transport systems across membranes, and the protection and stabilization of cell structures from damage by reactive oxygen species, along with fatty acid metabolism enzymes, proteinase inhibitors, ferritin and lipid-transfer proteins, and others functioning in the protection of macromolecules such as proteins and DNA from degradation (e.g., LEA proteins, osmotins, and chaperones) [1,4,37,38]. Our identification of varied DEGs common to during both CA, from early to late CA, and DA, from late CA to late DA, will be useful in unraveling the molecular mechanisms underpinning the responses to cold stress.

## Figures and Tables

**Figure 1 genes-11-00611-f001:**
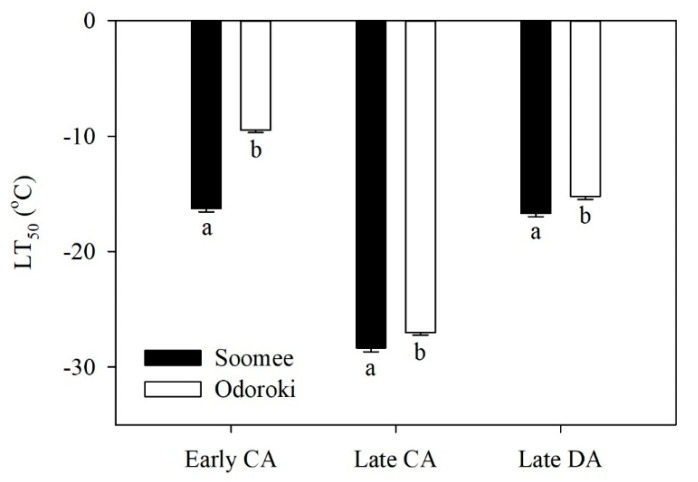
Seasonal changes in LT_50_ values determined by electrolyte leakage analysis of the ‘Soomee’ and ‘Odoroki’ peach tree shoots during cold acclimation (CA), from early to late CA, and deacclimation (DA), from late CA to late DA. Vertical bars are the standard errors of the means (*n* = 30). Different letters indicate significant differences within the same physiological stages using the Student’s *t*-test at *p* < 0.01.

**Figure 2 genes-11-00611-f002:**
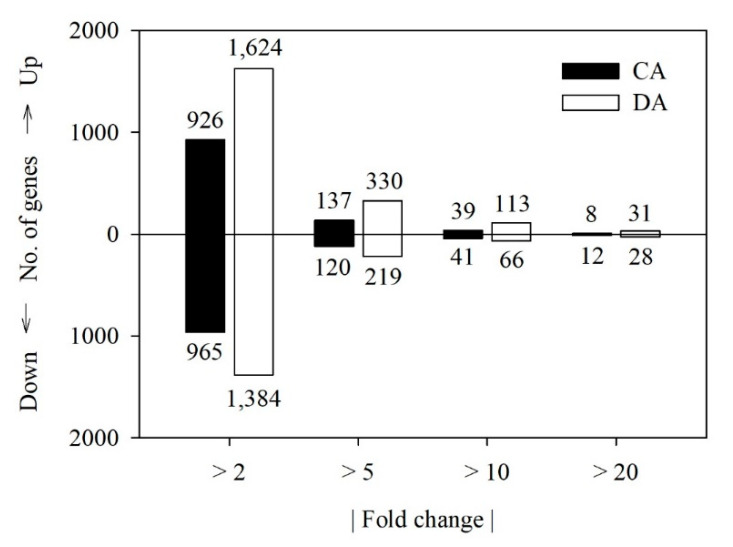
Number of up- and down-regulated genes with a |fold change| > 2 (*p* < 0.05) in the ‘Soomee’ peach tree shoots during cold acclimation (CA) and deacclimation (DA). Fold changes during CA and DA were determined based on the FPKM values at late CA and late DA as compared to those at early and late CA, respectively.

**Figure 3 genes-11-00611-f003:**
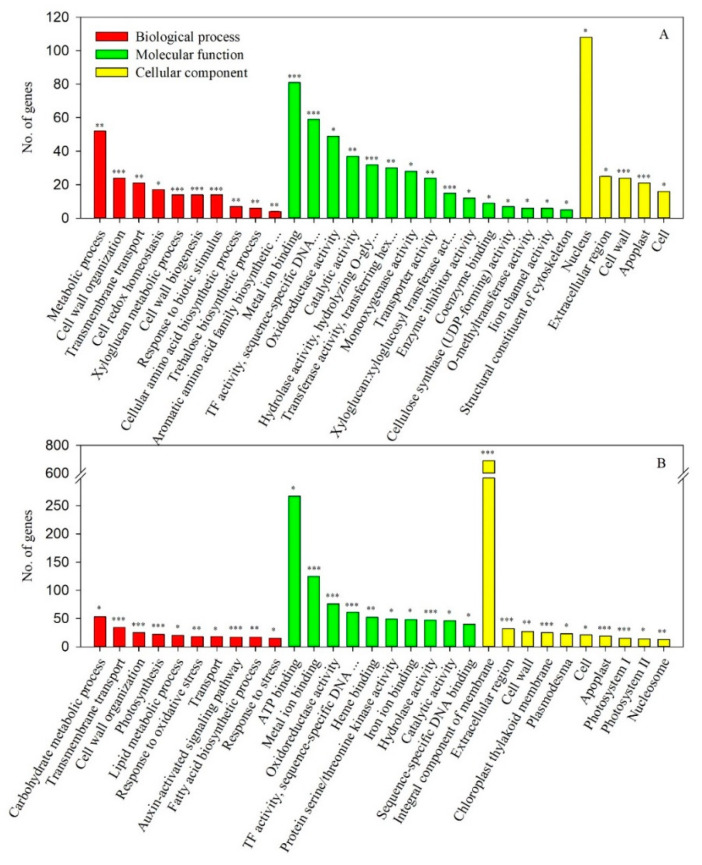
Gene ontology (GO) terms enriched DEGs with a |fold change| > 2 (*p* < 0.05) in the ‘Soomee’ peach tree shoots during cold acclimation (CA) (**A**) and deacclimation (DA) (**B**). GO terms were classified into biological process, molecular function, and cellular component categories. Fold changes were determined based on the FPKM values at late CA and late DA as compared to those at early and late CA, respectively. Asterisks (*, **, ***) on the bars indicate significance using the modified Fisher’s exact test at *p* < 0.05, 0.01, or 0.001, respectively. The significantly overrepresented DEGs included in each GO term are listed in Appendix A.

**Figure 4 genes-11-00611-f004:**
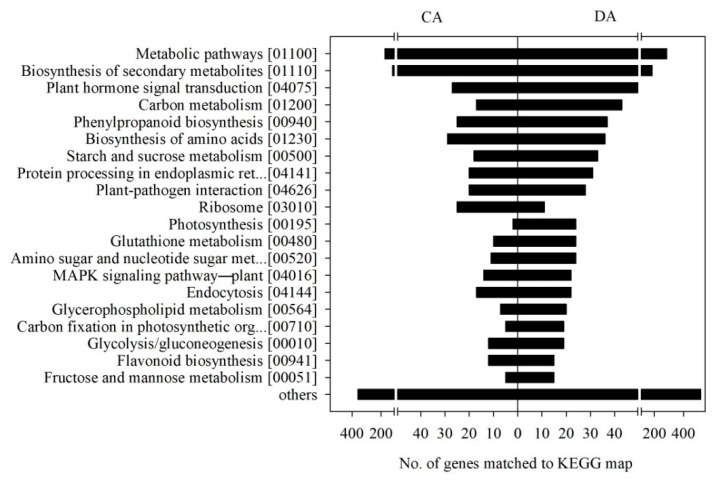
Biochemical pathways with their identification numbers, based on the KEGG database categories, enriched in the DEGs with a |fold change| > 2 (*p* < 0.05) in the ‘Soomee’ peach tree shoots during cold acclimation (CA) and deacclimation (DA). Fold changes were determined based on the FPKM values at late CA and late DA as compared to those at early and late CA, respectively.

**Figure 5 genes-11-00611-f005:**
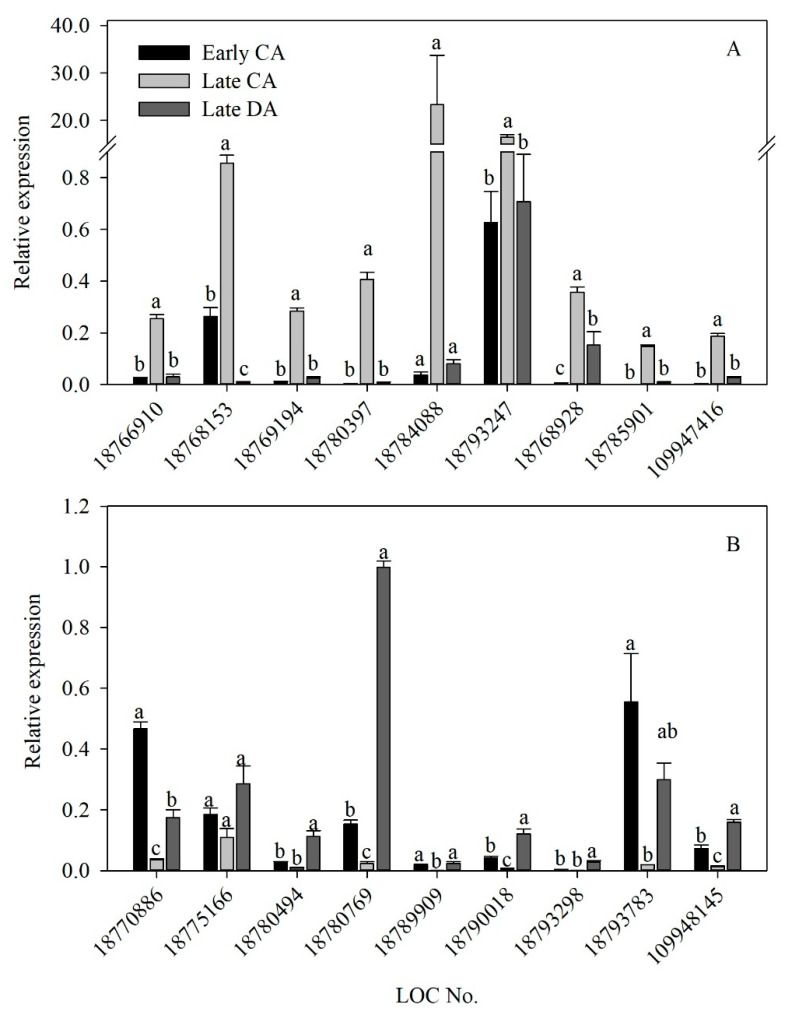
Relative gene expressions for the validation of (**A**) the selected nine DEGs up- and down-regulated during cold acclimation (CA), from early to late CA, and deacclimation (DA), from late CA to late DA, respectively, and (**B**) the selected nine DEGs down- and up-regulated during CA, from early to late CA, and DA, from late CA to late DA, respectively, in the ‘Soomee’ peach tree shoots. The DEGs were selected based on the significance of fold change during CA and DA. Vertical bars are the standard errors of the means (*n* = 3). Different letters indicate significant differences among the three physiological stages within the same DEGs using the Duncan’s multiple range test at *p* < 0.05. LOC18766910, *non-symbiotic hemoglobin*; LOC18768153, *squalene monooxygenase*; LOC18769194, *polygalacturonase inhibitor*; LOC18780397, *tonoplast dicarboxylate transporter*; LOC18784088, *late embryogenesis abundant protein 2*; LOC18793247, *extracellular ribonuclease LE*; LOC18768928, *bidirectional sugar transporter SWEET1*; LOC18785901, *14 kDa proline-rich protein DC2.15*; LOC109947416, *pEARLI1-like lipid-transfer protein 3*; LOC18770886, *O-acyltransferase WSD1*; LOC18775166, *phosphoenolpyruvate carboxykinase* (*ATP*); LOC18780494, *protein SRG1*; LOC18780769, *protein E6*; LOC18789909, *non-specific lipid-transfer protein 2*; LOC18790018, *aspartyl protease AED3*; LOC18793298, *cyclin-D1-1*; LOC18793783, *heterogeneous nuclear ribonucleoprotein A2 homolog 1*; LOC109948145, *vinorine synthase-like*.

**Figure 6 genes-11-00611-f006:**
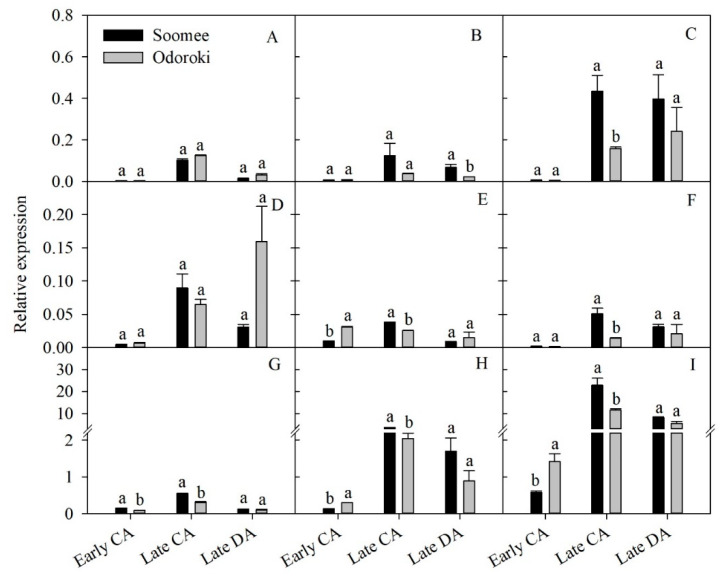
Comparative gene expressions of the selected nine DEGs up- and down-regulated during cold acclimation (CA), from early to late CA, and deacclimation (DA), from late CA to late DA, respectively, between the ‘Soomee’ and ‘Odoroki’ peach tree shoots. The DEGs were selected based on the significance of fold change during CA and DA. Vertical bars are the standard errors of the means (*n* = 3). Different letters indicate significant differences within the same physiological stages using the Student’s *t*-test at *p* < 0.05. (**A**), *non-symbiotic hemoglobin* (LOC18766910); (**B**), *squalene monooxygenase* (LOC18768153); (**C**), *polygalacturonase inhibitor* (LOC18769194); (**D**), *tonoplast dicarboxylate transporter* (LOC18780397); (**E**), *late embryogenesis abundant protein 2* (LOC18784088); (**F**), *extracellular ribonuclease LE* (LOC18793247); (**G**), *bidirectional sugar transporter SWEET1* (LOC18768928); (**H**), *14 kDa proline-rich protein DC2.15* (LOC18785901); (**I**), *pEARLI1-like lipid-transfer protein 3* (LOC109947416).

**Table 1 genes-11-00611-t001:** Transcriptome sequencing data statistics from the ‘Soomee’ peach tree shoots during cold acclimation (CA), from early to late CA, and deacclimation (DA), from late CA to late DA.

Physiological Stage	Total Read	Total Base	GC Content (%)	Q30 (%)	Mapping Ratio (%)
Early CA	66,451,463	6,683,605,100	46.3	95.4	96.9
Late CA	54,420,355	5,463,673,888	45.8	93.9	96.7
Late DA	50,845,192	5,122,738,506	45.6	96.3	97.2

Average read length was 101 bp. Each value is the mean of three biological replicates.

**Table 2 genes-11-00611-t002:** Correlations of the cold hardiness (LT_50_), and relative expression (RE) and log_2_(FPKM + 1) values of the selected DEGs during cold acclimation (CA), from early to late CA, and deacclimation (DA), from late CA to late DA, in the ‘Soomee’ peach tree shoots for the validation of the in silico data by qPCR.

Gene Symbol (LOC No.)	Fold Change		Correlation Coefficient (*r*)		
	CA	DA	LT_50_ and log_2_(FPKM + 1)	log_2_(FPKM + 1) and RE	RE and LT_50_
18766910	61.9 *	−18.7 *	−0.81 ^ns^	0.84 ^ns^	−0.99 *
18768153	12.4 *	−7.1 **	−0.62 ^ns^	0.86 ^ns^	−0.92 ^ns^
18769194	47.3 **	−2.4 ^ns^	−0.98 ^ns^	0.97 ^ns^	−0.99 *
18780397	20.9 *	−6.7 **	−0.99 *	0.99 ^ns^	−0.99 *
18784088	6.0 **	−53.4 **	−0.90 ^ns^	0.93 ^ns^	−0.99 ^ns^
18793247	20.6 *	−14.8 **	−0.96 ^ns^	0.94 ^ns^	−0.99 *
18768928	3.1 *	−93.7 **	−0.69 ^ns^	0.90 ^ns^	−0.93 ^ns^
18785901	13.5 *	−58.7 *	−0.96 ^ns^	0.96 ^ns^	−0.99 *
109947416	16.4 **	−7.5 ^ns^	−0.94 ^ns^	0.95 ^ns^	−0.99 *
18770886	−10.3 *	5.7 **	0.99 ^ns^	0.87 ^ns^	0.79 ^ns^
18775166	−6.2 **	15.6 **	0.90 ^ns^	0.96 ^ns^	0.77 ^ns^
18780494	−6.2 **	15.2 **	0.89 ^ns^	0.88 ^ns^	0.58 ^ns^
18780769	−9.5 *	32.5 **	0.91 ^ns^	0.83 ^ns^	0.53 ^ns^
18789909	−16.7 *	63.4 **	0.91 ^ns^	0.99 **	0.91 ^ns^
18790018	−6.8 *	12.7 **	0.94 ^ns^	0.87 ^ns^	0.67 ^ns^
18793298	−5.2 *	27.0 **	0.80 ^ns^	0.91 ^ns^	0.50 ^ns^
18793783	−52.2 **	16.3 **	0.98 ^ns^	0.97 ^ns^	0.91 ^ns^
109948145	−7.5 **	9.7 **	0.97 ^ns^	0.87 ^ns^	0.75 ^ns^

^ns,^ *^,^ ** not significant or significant at *p* < 0.05 or 0.01, respectively, using the Student’s *t*-test.

**Table 3 genes-11-00611-t003:** Top 20 common DEGs up-regulated during cold acclimation (CA), from early to late CA, and down-regulated during deacclimation (DA), from late CA to late DA, in the ‘Soomee’ peach tree shoots.

Gene Symbol (LOC No.)	Gene Description	Fold Change (Expression Volume)		Correlation Coefficient (*r*)
		CA	DA	
18792263	Early light-induced protein 1, chloroplastic	15.7 (10.3) *	−53.0 (9.4) *	−0.92 ^ns^
18785901	14 kDa proline-rich protein DC2.15	13.5 (8.3) *	−58.7 (6.9) *	−0.90 ^ns^
18784998	21 kDa protein	9.0 (8.6) **	−5.6 (9.0) **	−0.99 ^ns^
18788919	F-box/Kelch-repeat protein At1g67480	6.9 (7.3) **	−7.0 (7.3) **	−0.99 ^ns^
18789866	Chaperone protein dnaJ 11, chloroplastic	8.9 (6.0) **	−16.2 (5.4) **	−0.97 ^ns^
18767775	Indole-3-acetic acid-induced protein ARG2	3.9 (7.8) *	−12.8 (6.8) *	−0.83 ^ns^
18778639	Stem-specific protein TSJT1	11.7 (5.3) **	−20.0 (4.8) *	−0.98 ^ns^
18785018	Glutamate dehydrogenase 2	11.9 (5.3) **	−9.4 (5.6) **	−0.99 *
18766910	Non-symbiotic hemoglobin	61.9 (2.9) *	−18.7 (4.6) *	−0.96 ^ns^
18768928	Bidirectional sugar transporter SWEET1	3.1 (7.2) *	−93.7 (3.6) **	−0.62 ^ns^
18780397	Tonoplast dicarboxylate transporter	20.9 (4.3) *	−6.7 (5.5) **	−0.94 ^ns^
18793304	GEM-like protein 5	2.6 (7.3) *	−16.1 (5.8) **	−0.69 ^ns^
18768308	Protein EXORDIUM-like 2	4.7 (7.2) *	−5.3 (7.1) **	−0.99 ^ns^
18767434	Glutaredoxin-C11	5.5 (5.7) *	−42.7 (3.5) **	−0.82 ^ns^
18782474	Triacylglycerol lipase 2	5.7 (6.3) **	−5.7 (6.4) **	−0.99 *
18784501	ABC transporter C family member 8	20.7 (4.1) *	−5.0 (5.6) *	−0.90 ^ns^
18792321	Probable xyloglucan endotransglucosylase/hydrolase protein 23	2.0 (8.8) *	−7.1 (7.8) *	−0.70 ^ns^
18787602	Sorbitol dehydrogenase	4.6 (7.6) **	−3.2 (7.9) **	−0.98 ^ns^
18770266	LOW QUALITY PROTEIN: protein PIN-LIKES 7	26.9 (3.6) **	−5.4 (5.3) *	−0.89 ^ns^
18792372	Zinc finger protein ZAT10	8.2 (5.1) **	−7.0 (5.2) **	−0.99 *

Genes were ranked based on fold change and expression volume (geometric mean of FPKM values) of DEGs between the physiological stages. Correlation coefficients were determined between cold hardiness (LT_50_) and log_2_(FPKM + 1) of the selected DEGs. ^ns^, *, ** not significant or significant at false discovery rate < 0.05 or 0.01, respectively, using the Benjamini–Hochberg method.

**Table 4 genes-11-00611-t004:** Sequences of forward and reverse primers used for qPCR of the selected nine DEGs up- and down-regulated during cold acclimation (CA), from early to late CA, and deacclimation (DA), from late CA to late DA, respectively, and the selected nine DEGs down- and up-regulated during CA, from early to late CA, and DA, from late CA to late DA, respectively, in the ‘Soomee’ peach tree shoots.

Gene Symbol (LOC No.)	Gene Description	Primer Sequence
DEGs up- and down-regulated during CA and DA, respectively		
18766910	Non-symbiotic hemoglobin	F: AGAGCAGGAAACATTGGTGG R: CTTCTGAGCTGATGGTGCAA
18768153	Squalene monooxygenase	F: CCAAGCATGCTTCGACTACT R: TCAGTCTAGCTCCAACCCAA
18769194	Polygalacturonase inhibitor	F: ACCATCCTAAACCCAGCTCT R: AGTACCAGTCACAGCAGTCT
18780397	Tonoplast dicarboxylate transporter	F: ATGGATGATGTCATTGCGCT R: CTCCGCAGAATAGCATGGTT
18784088	Late embryogenesis abundant protein 2	F: AAACAGAGGGCAGAAGAAGC R: CTTTTCACAGCATCAGCAGC
18793247	Extracellular ribonuclease LE	F: GAATGGGAAAAGCATGGCAC R: TGGTTGTATGCCTGCACTTT
18768928	Bidirectional sugar transporter SWEET1	F: CCTTTTGTTGCTGTGCCAAA R: TTCATCAGCAGTAGCAGCAG
18785901	14 kDa proline-rich protein DC2.15	F: CACCAACACCACCAAAAACC R: TAGGGTCGAAGGAGGTTGTT
109947416	pEARLI1-like lipid-transfer protein 3	F: TGATGCTGCTGTGTGTCTTT R: GCATTGGAAGTCTGTTGGGA
DEGs down- and up-regulated during CA and DA, respectively		
18770886	*O*-Acyltransferase WSD1	F: TCTCAGGTTTCTGCAACACC R: AATGTATTTCCCGCATGCCT
18775166	Phosphoenolpyruvate carboxykinase (ATP)	F: ATTCTGGCTGCAATATGGGG R: CAGCATCTTATCAGCCAGCA
18780494	Protein SRG1	F: AGACGTGGAAGGTTTTGGAC R: ACTGGACCAATTTCACCACC
18780769	Protein E6	F: CACCACCAACAACAACAACC R: TGGCACGAACTCTTCTTCAC
18789909	Non-specific lipid-transfer protein 2	F: TTCTTGTCTTTGGGGGAAGC R: AAATGGGGAGCCACAAGTG
18790018	Aspartyl protease AED3	F: GCATCAATTCCATCTCGGGT R: AAGCTGTTCTCCAATGGCAA
18793298	Cyclin-D1-1	F: ATGTCATCGGTTTGGAGCTC R: CCTCCAATCCAAAACACCCA
18793783	Heterogeneous nuclear ribonucleoprotein A2 homolog 1	F: GCCGATGTAGTGACTTTTCCT R: GGTTCTTGTTCCCGTTCAGT
109948145	Vinorine synthase-like	F: ACTGCTTGCTATCTGACGTG R: CCATCCTGGTCCGTAAGTTG

F, forward; R, reverse.

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
