# Peer review of "Transcriptome Analysis of Genes Involved in Cold Hardiness of Peach Tree (Prunus persica) Shoots during Cold Acclimation and Deacclimation"

_genes, 2020, doi:10.3390/genes11060611_

Round 1

Reviewer 1 Report

Comments for the authors,

This manuscript demonstrated that how genes were expressed during cold acclimation and deacclimation of peach tree by RNA-seq. To identify new genes involved in cold acclimation and de-acclimation, the authors analyzed differentially expressed genes (DEGs) and the correlation between expression patterns and cold tolerance. This reviewer thinks that the most important point of this study is using samples of peach trees in field conditions. Unlike the experimental conditions, it is difficult to analyze field sample because field conditions are not stable and several stresses are intricately intertwined. However, this is natural conditions and this kind of study is very important for understanding the actual response of plants during cold acclimation and deacclimation.

Although this study is important and interesting, this reviewer realized that there are some major points regarding the description or definition of samples for analyses. Major and minor points include the following:

Major points,

1.

In abstract, page 1, line 14-15, the authors described that “Cold hardiness of the shoots increased during CA, from early CA (end of October) to late CA (middle of January), but decreased during DA, from late CA to late DA (middle of March), as indicated by decreased and increased LT50 values, respectively.” and in materials and methods, page 2, line 74-77, “The shoot internodes of the ‘Soomee’ and ‘Odoroki’ peach trees were separately collected at early CA (end of October), late CA (middle of January), and late DA (middle of March), immediately frozen in liquid nitrogen, and stored at –80 °C until use.”. The samples for this study was “early CA”, “late CA” and “late DA”. However, analysis after cold hardiness test and RNA-seq, categorized samples for analyses were “during CA or during DA”, for example, in page 5, line 184-185, “In total, 1,891 and 3,008 transcripts were differentially expressed during CA and DA, respectively, with a |fold change| > 2 (P < 0.05) (Figure 2).” What this reviewer is concerned about is whether the CA sample under analysis is "early CA" or "late CA"?

And in page 5, line 193-195, Legends of Fig.2, “Fold changes were determined based on the FPKM values at the end of CA and DA as compared to those at the beginning of CA and DA, respectively.” What is “end of CA”, or “beginning of CA and DA”. The authors used different words for each sample and it is hard to tell which sample they are referring to. The authors should use unifying words.

2.

In page 5, line 185-187, the authors described that “The number of significant DEGs was higher during DA than during CA, which indicates that DA is not merely a reverse of the CA process, but a separate physiological stage preparing trees for the resumption of growth and development.” This reviewer thinks that this is because the samples compared in this experiment are early CA and late CA. If this is a comparison with non-acclimation, the DEGs might be increased.

And also, in page 7, line 240-242, “DREB1E, DREB2C, and DREB2D were significantly up- and down-regulated during CA and DA, respectively, while DREB1A and DREB3 were significantly down-regulated during both CA and DA (Figure S2).” As shown in Figure S2, the expression of DREB TFs were already up-regulated in early CA. This reviewer thinks that this data may indicate that early CA is already cold acclimated compared with real non-acclimated samples because some DREB genes already expressed in early CA. In other words, the low temperature response has already begun in early CA, and there is no non-acclimation as a control in this study. The authors should consider the definition and control of the samples for analysis in this study.

3.

In page 14, line 403-405, the authors described that “According to the in silico data, the dehydration-responsive genes mentioned above were up-regulated during early CA, maintained at their high expression levels during CA, and were then greatly down-regulated during DA (Figure S3).” It seems that early CA is already cold acclimated, and not a non-acclimated sample. When non-acclimated sample is taken as a control, it is possible that LEAs may come up for candidates and respond similarly to CA processes in other plants. This paper is a transcriptome of an analysis of what happens in late CA and DA when early CA is used as the control. If the authors don't describe this clearly, it would lead to the conclusion that LEA is not an important gene that is not in the top 20 in peach CA. The authors should clarify what happened to the late CA, DA in the field when the early CA was used as a control sample.

4.

In introduction, the authors mentioned that transcriptome analyses using grape and peach have been reported in other studies. If the data obtained in this study can be compared with other transcriptome data, this reviewer thinks that this paper can be improved by considering new differences between the data obtained in the laboratory and the data obtained in the field.

Minor points,

5.

Figure 7 shows what appears to be the same data as in table 3.

What's the difference? What has been revealed with this.

The authors should make this clear.

6.

In abstract, page 1, line 31-32, “These DEGs included early light-induced protein 1 (chloroplastic), 14-kDa proline-rich protein DC2.15, glutamate dehydrogenase 2, and triacylglycerol lipase 2.” DEGs are positively correlated with phenotypes, but these genes are only candidates. This reviewer thinks that a little more concrete evidence is needed to conclude this sentence at the end of the abstract.

7.

In page 5, line 183, ”Of the 18,344 assembled transcripts,”. Suddenly, the number, 18,344, came up. The method of calculating the assembled transcripts is described in materials and methods, but this reviewer thinks that a one-sentence explanation is needed before this.

8.

In Figure 5, the genes cannot be distinguished in grayscale and are difficult to understand. It is also necessary to create a graph for each gene in order to see the increase or decrease in expression between samples of each gene, rather than a comparison between genes.

For example, make it a line graph and color it. Alternatively, the graph may be divided by gene. If it is divided by gene, it can be understood in grayscale.

9.

In page 13, line 354-356, in legends of Figure 6, “Comparative gene expressions of the selected nine DEGs up- and down-regulated during cold acclimation (CA) and deacclimation (DA), respectively, and the selected nine DEGs down- and up-regulated during CA and DA, respectively, between the ‘Soomee’ and ‘Odoroki’ peach tree shoots.”

In this study, "and the nine selected DEGs were down-regulated and up-regulated during CA and DA, respectively," but in Figure 6, this type of gene was not analyzed. The authors should correct it if this is a mistake.

Author Response

  1. Samples for comparison
    Peach tree shoots under field condition were separately collected at the end of October, middle of January, and middle of March, which represent early CA, late CA, and late DA, respectively, for cold hardiness determination and transcriptome analysis. Thus, words describing the sample for comparison have now been unified to early CA, late CA, and late DA, when they are appropriate, throughout the manuscript.
  2. DEG analysis in comparison with non-acclimated samples
    DEG analyses during CA and DA were performed by paired comparisons between the FPKM values at early CA and late CA, and between those at late CA and late DA, respectively. Thus, non-acclimated samples were not included as controls. This fact has already been described on lines 148-150.
  3. Up-regulated DEGs at early CA
    Some DEGs up-regulated at early CA through late CA might not be selected as candidate genes determining cold hardiness. If their expressions were significantly reduced during DA, the DEGs are thought to be associated with cold hardiness.
  4. New difference between the data obtained in the laboratory and in the field
    Most previous transcriptome analyses in peach trees have been performed under specific temperature conditions in the laboratory. In the present study, however, we analyzed the seasonal changes of the transcriptomes in the field. This fact has already been addressed in Introduction on lines 62-65.
  5. Figure 7
    As pointed out by the reviewer, the data for Figure 7 were from Table 3. Thus, Figure 7 has now been deleted.
  6. Last sentence in the Abstract
    Since the DEGs are only candidates, the last sentence in the Abstract has been changed to ‘These DEGs, including early light-induced protein 1 (chloroplastic), 14-kDa proline-rich protein DC2.15, glutamate dehydrogenase 2, and triacylglycerol lipase 2, could be candidate genes determining cold hardiness’ on lines 31-33.
  7. Explanation for calculating the assembled transcripts
    The transcripts expressed in any shoot samples examined were counted to be 18,344. However, the numbers of the transcripts identified in all the samples examined were 17,208 following the removal of the transcripts showing zero FPKM in any samples. This fact has been described in the section 2.5 on lines 143-146.
  8. Figure 5
    Figure 5 has been redrawn for each DEG for clearly showing the seasonal changes of their relative expression.
  9. Legends for Figure 6
    We made a mistake on this. The words ‘the selected nine DEGs down- and up-regulated during CA and DA, respectively’ have now been deleted from the legends for Figure 6.

Reviewer 2 Report

The manuscript titled “Transcriptome Analysis of Genes Involved in Cold Hardiness of Peach Tree (Prunus persica) Shoots during Cold Acclimation and Deacclimation” by Yu et al. describes transcriptomic changes that take place in peach tree shoots after cold acclimation and deacclimation processes. In base of that, they select some genes likely involve in them and compare their expression patterns with another peach tree cultivar that differs in cold hardiness. The results are interesting and the manuscript is very well written. However, in my opinion, some questions have to be addressed before publication in this journal:

  • Lines 72-78. Authors must explain more in detail the experimental design. How many trees they sampled? How many shoots they analyzed per condition? What they consider a replicate? Are they biological replicates?
  • Lines 141-144. Could authors specify here how the calculate CA and DA DEGs?
  • Line 195. Authors specify beginning of DA. Is that mean late CA condition?
  • Line 214. Authors indicate “genes involved in altering secondary metabolic pathways during CA.” Could they give some examples of those genes?
  • Lines 243-244. “In the cellular component category, the GO term ‘nucleus’ was dominant among the DEGs (Figure 3A); other enriched terms included ‘extracellular region’, ‘cell wall’, and ‘apoplast’.” Authors could relate cellular component ‘nucleus’ with ‘DNA binding’ molecular function discussed above, and ‘cell wall’ with the biological processes ‘cell wall organization’ and ‘cell wall biogenesis’ that also are enriched.
  • Line 267-269. “These GO terms suggest that the expression of genes involved in the metabolic processes preparing for the resumption of growth and development is induced as cold hardiness decreases during DA.” Could they give some examples of those genes?
  • Line 348. There is another exception: D, tonoplast dicarboxylate transporter (LOC18780397) does not present significant differences.
  • Authors may draw main conclusion of their study at the end of the manuscript.
  • In my opinion authors should prepare as supplementary material a file with the genes included in each GO term significantly overrepresented. It would help to understand their present study and also could be very valuable for another authors working on cold stress.

Author Response

  1. Details in experimental design
    Shoots were collected separately from three different trees at each physiological stage to provide three biological replicates. This fact has now been described in the section 2.1 on lines 81-83.
  2. Calculation of CA and DA DEGs
    DEG analyses during CA and DA were performed by paired comparisons between the FPKM values at early CA and late CA, and between those at late CA and late DA, respectively. CA and DA DEGs were selected based on the significance of fold changes between early CA and late CA and between late CA and late DA, respectively. The fold changes were calculated as the ratios between squared values of normalized FPKM means at each stage, as described on lines 150-153.
  3. Beginning of DA in the legends for Figure 2
    The words describing the sample for comparison have now been unified to early CA, late CA, and late DA throughout the manuscript. The legends for Figure 2 have been corrected accordingly and those for Figures 3 and 4 have also been corrected.
  4. Example of the genes involved in altering secondary metabolic pathway during CA
    For examples, squalene monooxygenase (LOC18768153), beta-glucosidase 12 (LOC18770931), and adenylate isopentenyltransferase 3, chloroplastic (LOC18766570) could be included as up-regulated genes, while isoleucine N-monooxygenase 2 (LOC18775172), polyphenol oxidase, chloroplastic (LOC18778195), and gibberellin 2-beta-dioxygenase (LOC18779667) could be included as down-regulated genes. This fact has been described on lines 226-230.
  5. Relations of GO terms with the enriched terms
    GO terms among the biological process, molecular function, and cellular component categories are difficult to be related. Instead, the GO term enrichment within each category has already been described in the section 3.4.
  6. Example of the genes involved in the metabolic processes preparing for the resumption of growth and development
    The genes encoding 1-aminocyclopropane-1-carboxylate oxidase 1 (LOC18781368), leucoanthocyanidin reductase (LOC18789589), peroxidases (LOC18773443, LOC18769960, and LOC18772634), endoglucanase 9 (LOC18768088), and phosphoenolpyruvate carboxykinase (ATP) (LOC18775166) were significantly up-regulated only during DA. These genes might be involved in the metabolic processes preparing for the resumption of growth and development. This fact has been described on lines 285-289.
  7. Tonoplast dicarboxylate transporter in Figure 6D
    The DEG encoding tonoplast dicarboxylate transporter (LOC18780397) did not show significant differences at three different physiological stages. This fact has now been added into the sentence on lines 367-370.
  8. Main conclusion
    Actually, the third paragraph in the section 3.8 was the main conclusion of this study. The third and the fourth paragraphs have now been switched. Reference numbers have been changed in the text and References accordingly.
  9. Supplementary materials
    Information on the significantly overrepresented DEGs included in each GO term has been provided as an Excel file in Table S1. This fact has now been described on lines 221-222. In addition, the legends for Table S1 have been written on lines 419-422.

Reviewer 3 Report

In the paper “Transcriptome Analysis of Genes Involved in Cold 2 Hardiness of Peach Tree (Prunus persica) Shoots 3 during Cold Acclimation and Deacclimation”, Yu and colleagues analyzed the transcriptomes in the shoots of ‘Soomee’ peach trees (Prunus persica) during cold acclimation (CA) and deacclimation (DA) to identify genes involved in cold hardiness. Authors also try to compare gene expression patterns of this genotype with those in the shoots of ‘Odoroki’ peach trees, which is less cold-hardy than ‘Soomee’.

The work is well written and properly presented.

However, I have a few quite relevant concerns.

  • Numerous works have been carried out on the transcriptome of dormant buds in woody plants, in order to understand what are the adaptation mechanisms in this organ so precious and fragile (i.e. https://doi.org/10.3389/fpls.2020.00180), but also in bark, stigma and leaf, as well documented by the manuscript’s references. However, in this case, authors focus on the transcriptional changes occurring in the shoot (internodes, as specify in the Material & Methods section).

This could represent the originality of the work, but, in this case, it must be highlighted and specified. The terms "deacclimation" and "acclimation" could recall the bud, which, if I understand correctly, is excluded from the analysis. On the other hand, in line 81: bud-attached shoot. Did authors use bud-attached shoots for determining the levels of cold hardiness and internode for transcriptomics?

This point needs to be clarified.

In any case, authors should specify the amount of each sample. The representativeness of the sample is very important.

  • The second main concern is about the involvement of ‘Odoroki’ genotype. In my opinion, the idea of ​​a genotype with a different cold tolerance is very interesting, but the analysis of the latter seems to be reduced to Figure 6, barely commented. I think that the authors should valorize this aspect.

Moreover, why just the relative expressions of the nine DEGs up-regulated during CA and down-regulated during DA were analyzed in ‘Odoroki’?

  • It is not clear the relevance of the last paragraph. Also in this case I think the authors should valorize this analysis. In addition, a clarification is needed in order to elucidate the relationship between the nine DEGs selected in paragraph 3.8 and the 20 in 3.9. Otherwise, they can appear conflicting.

Minor points:

  • Line 59: Jia is Jiao
  • Lines 349-352: generic and not very supported sentence
  • A more comprehensive discussion of the findings will greatly improve the manuscript

Author Response

  1. Shoot samples
    The internodes of peach tree shoots were used for both cold hardiness determination and transcriptome analysis. During the cold hardiness determination, however, the bud-attached shoot internodes were used for freezing and thawing in a programmable bath circulator to avoid the damage caused by bud separation from the shoots. This fact has been described in the section 2.1 on lines 76-81.
  2. Analysis of ‘Odoroki’ cultivar
    Transcriptomes were analyzed in the shoots of cold-hardy peach cultivar ‘Soomee’ at three different physiological stages. DEGs associated with cold hardiness were selected based on the correlation between their relative expression and LT50 values. The DEG expressions were compared with those of less cold-hardy cultivar ‘Odoroki’ for confirming the involvement of DEGs in cold hardiness. In this way, the efforts for identifying the DEGs could substantially be reduced.
  3. Relevance of last paragraph
    Actually, the third paragraph in the section 3.8 was the main conclusion of this study. The third and the fourth paragraphs have now been switched. Reference numbers have been changed in the text and References accordingly.
  4. Relationship between the nine DEGs selected in the section 3.8 and the top 20 DEGs
    The nine DEGs were selected based on the significance of fold changes between early CA and late CA and between late CA and late DA. The DEG identification results were validated against the nine DEGs by performing RT-qPCR. However, the top 20 DEGs up-regulated between early CA and late CA and down-regulated between late CA and late DA were selected based on the expression volume (geometric mean of FPKM values) as well as the fold change. Thus, some of the nine DEGs were not included in the top 20 DEGs. In addition, statistically significant differences among means in Table 3 were determined using the Benjamini-Hochberg method at false discovery rate < 0.05 or 0.01. This fact has been added in the section 2.7 on lines 171-172.
  5. Author name
    The author name of ‘Jia’ on line 60 has been corrected to ‘Jiao’.
  6. Sentence on lines 349-352
    As pointed out by the reviewer, the sentence is generic and not supported. Thus, the sentence has now been deleted from the line 370 in the revised version.
  7. More comprehensive discussion
    Discussion has been improved during the revision of the manuscript.

Round 2

Reviewer 1 Report

1.

The authors answered in response #1 that “Thus, words describing the sample for comparison have now been unified to early CA, late CA, and late DA, when they are appropriate, throughout the manuscript.” But still in the text of manuscript, many “during CA” are still remaining.

This reviewer thinks that “during CA” means that changing process compared with non-acclimation during cold temperature stress. In the response #2, the authors answered that “DEG analyses during CA and DA were performed by paired comparisons between the FPKM values at early CA and late CA, and between those at late CA and late DA, respectively. Thus, non-acclimated samples were not included as controls.” It means that “during CA” is changes late CA compared with early CA in this manuscript.

Revised manuscript changed that description only in figure legends. It might make readers misread of definition of “CA”. The authors should correct all of the description about “CA”, if the meaning is “changes from early CA to late CA”.

2.

The authors answered in response #7 and add the description of calculation in the section 2.5 on lines 143-146. These 2 sentences are results and should move to result section 3.3.

Author Response

  1. Samples for comparison
    Peach tree shoots under field condition were separately collected at the end of October, middle of January, and middle of March, which represent early CA, late CA, and late DA, respectively, for cold hardiness determination and transcriptome analysis. Thus, words describing the sample for comparison have now been unified to early CA, late CA, and late DA in the text as well as in Table and Figure legends.
  2. Descriptions of calculating the assembled transcripts
    The descriptions of calculating the assembled transcripts have been moved from the section 2.5 to the section 3.3 on lines 196-198.
  3. Other minor corrections have been made accordingly.

Reviewer 3 Report

I notice a great effort in improving the manuscript. I really appreciate it.

I don't want to be obstinate, but, in my opinion, a more precise description of "Plant material" section could be useful. I.e. how many shoots for each replicate?

Moreover, with the term "internode" I mean a part of a plant stem between two of the nodes. I understand that cold hardiness has been measured on buds, but I'm still not sure if buds are included in transcriptomic analysis.

I recognize this should be considered marginal, but I think is important, for the discussion about genes and gene functions, understanding where the results come from (tissue type).

Author Response

  1. Number of shoot samples
    Six and three shoots from each tree were used for cold hardiness determination and transcriptome analysis, respectively, at each physiological stage. This fact has been described in section 2.1 on lines 81-83.
  2. Shoot samples for cold hardiness determination and transcriptome analysis
    For cold hardiness determination, bud-attached shoots in 8-cm pieces were subjected to freezing treatment because damage caused by bud separation from the shoots may interfere with the determination. Following the freezing treatment, the internodes from the bud-attached shoots were cut into 1-cm pieces, and then incubated for allowing electrolyte leakage.
    For transcriptome analysis, total RNA was extracted from the shoot internodes, which had been immediately frozen in liquid nitrogen upon the collection from the field at each physiological stage and stored at –80℃.
    Therefore, no buds were included in either cold hardiness determination or transcriptome analysis. These facts have been described on lines 86-88, 95-97, and 107-108.
  3. Other minor corrections have been made accordingly.